# Targeting the TLK1-MK5 Axis Suppresses Prostate Cancer Metastasis

**DOI:** 10.3390/cancers17071187

**Published:** 2025-03-31

**Authors:** Damilola Olatunde, Omar Coronel Franco, Matthias Gaestel, Arrigo De Benedetti

**Affiliations:** 1Department of Biochemistry and Molecular Biology, Louisiana State University Health Shreveport, Shreveport, LA 71103, USA; damilola.olatunde@lsuhs.edu (D.O.); omar.francocoronel@lsuhs.edu (O.C.F.); 2Institute of Cell Biochemistry, Hannover Medical School, 30625 Hannover, Germany; gaestel.matthias@mh-hannover.de

**Keywords:** TLK1/1B, MAPKAPK5/MK5/PRAK, TRAMP mice, prostate cancer metastasis

## Abstract

We demonstrate with pharmacological and genetic approaches that by targeting the TLK1>MK5 nexus it was possible to strongly curtail the spread of metastatic, AR-independent PC3 cells and age-related metastases in a TRAMP/MK5-KO model. Importantly, one targeting drug (GLPG—MK5i) has been proven safe in Phase I clinical trials for the treatment of arthritis, while the other (J54—TLKi) is now commercially available and was shown to have no significant behavioral or fitness side effects.

## 1. Introduction

Prostate cancer (PCa) typically progresses slowly, especially when effectively managed with androgen deprivation therapy (ADT). However, metastases are responsible for approximately 30,000 deaths each year in the US. Although organ-confined PCa is clinically manageable with a nearly 100% five-year survival rate, metastatic PCa significantly reduces survival. Patients with liver, lung, bone, and lymph node metastases have life expectancies of about 14, 19, 21, and 32 months, respectively [1]. ADT is the standard treatment for PCa, but after 1–2 years, the cancer often relapses as *metastatic Castration Resistant PCa* (mCRPC), for which there is currently no cure. Cancer cells use two essential processes—motility and invasion—to detach from the primary tumor site and disseminate throughout the body [2,3]. The coordinated action of various factors regulates these processes, multiple signal transduction pathways, gene expression, cytoskeletal changes, and remodeling of extracellular matrices, all enabling the cells to invade and migrate into new tissues [3,4,5,6,7].

Tousled-like kinases (TLK) are serine/threonine protein kinases with well-established roles in DNA replication, transcription, chromosomal segregation, and DNA damage response and repair (rev. in [8,9]). However, TLK involvement in cellular motility is an under-studied area. The mammalian genome encodes two TLKs, *Tlk1* and *Tlk2*, which share 89% homology in their entire amino acid sequence and 94% similarity in their C-terminal kinase domains. TLK1 and TLK2 have partially redundant functions in the maintenance of genome stability [10]. Recently, two research groups independently reported that TLK2 enhances the migration rate and invasiveness of breast cancer and glioblastoma cell lines [11,12]. Another study revealed that *TLK* in *Drosophila* (only one *TLK* gene) is required for the collective migration of border cells by activating the JAK/STAT signaling pathway in the ovary, a phenomenon reminiscent of cancer migration and metastasis [13]. Our previous studies demonstrated that treatment with ADT/ARSI leads to higher translation of TLK1B mRNA in PCa via activation of the compensatory mTOR pathway [14]. Moreover, we observed genomic amplification and higher expression of TLK1 and one of its key downstream effector kinases—MK5—in metastatic PCa through our interrogation of TCGA, SU2C, and other public PCa patient databases [15]. This upregulation of TLK1/1B may mediate initial resistance to ADT and promote the migration of PCa cells from the primary tumor site and colonization of secondary locations.

In TRAMP mice (one of the best-studied genetic models of PCa development), expression of the PB-Tag is restricted to lobes of the prostate, and the temporal pattern of transgene expression correlates with sexual maturity [16,17,18]. TRAMP mice display high-grade PIN and/or well-differentiated PRAD by 10–12 weeks of age. Ultimately, TRAMP mice spontaneously develop invasive primary adenocarcinomas that routinely metastasize to the lymph nodes and lungs [18]. Early castration (12 weeks) results in a decrease in tumor volume burden but ultimately has no effect on time and progression to mCRPC and even the more aggressive form, NEPC [17], thus creating an opportunity to identify molecular targets and therapies relevant to delay or prevent progression to metastases and/or NEPC. While in vivo it has been difficult to monitor the progression of TRAMP tumors from localized lesions to distal metastases and the progression to castration resistance (or ARSI tolerance) directly, a careful study aimed at comparing progression in the TRAMP model with that of TRAMP-derived TC2 cells xenografted subQ in syngeneic C57BL/6 showed that the development of CRPC and resistance to enzalutamide (ENZ) was rapid and complete in two weeks [19].

We reported in several models that the expression of the TLK1B splice variant is increased via an mTOR-driven translational derepression following ADT/ARSI, which is consistent with results obtained with cell lines and many experiments (rev in [8,20]). One of the key interactors/substrates of TLK1/1B was identified as MK5/PRAK [15,21] in an important kinase relay (mTOR>TLK1B>MK5) that we have shown to impact motility, invasion, and metastatic spread of several PCa cell models [15].

The protein kinase MK5 (also known as PRAK, MAPKAPK5) shows significant sequence similarities to the stress-activated kinases MK2 and MK3, but the mechanism of its activation is completely different [22]. Most importantly, MK5 is found to interact with the atypical MAPKs ERK3 and ERK4, and it seems that resulting from this interaction, the partners mutually phosphorylate in trans and/or in cis [23,24,25,26]. Since the roles of ERK3 and ERK4 in cancer have not been clearly defined until now (e.g., see [27,28]), the role of MK5 in cancer is also far from being understood. The original findings that MK5/PRAK could act as a tumor suppressor [29], tumor promoter [30], or modulator of tumor metastasis [31] are obtained with a PRAK mouse “knock out” strain, which still expresses exon-deleted truncated MK5/PRAK mRNA and protein interfering with its functions [22,32]. Therefore, no clear evidence for these functions was given by these studies. However, a role for MK5 in the oncogenic Hippo-YAP pathway has been described recently where MK5 stabilizes YAP and promotes cancer [33], although we were not able to find any specific MK5 phosphorylation sites on YAP via an in vitro kinase assay using the recombinant proteins [34]. As mentioned above, an activating interaction between tousled-like kinase 1 (TLK1) and MK5 has been identified, increasing cell mobility and cancer metastasis [15,20]. Interestingly, in another report, MK5 was implicated in inactivating phosphorylation of Rheb [35] and thus priming the multifaceted action of mTOR that could also play a role in cell motility [36]. However, this report states p38β as a non-canonical activator of PRAK. It uses MEFs from the PRAK mouse “knock out” strain, which still expresses exon-deleted truncated MK5/PRAK mRNA and active protein. In summary, the role of MK5 in cancer progression or suppression has remained controversial. Hence, our current report sheds new light on the role of MK5 in metastasis. Here, we specifically decided to study the potential for a real-situation impact of this TLK1B>MK5 axis both via a pharmacologic approach and in a genetic model of spontaneous PCa development and metastatic progression by combining the TRAMP model with MK5-deficient mice (MK5-KO).

## 2. Results and Discussion

### 2.1. Interrogation of Expression Reports

It is noteworthy that the TLK1 gene was identified by co-expression analysis using WGCNA as a key driver of PCa with poor prognosis [37] but not followed systematically in mouse models of PCa; this is particularly evident for patients with low Gleason scores that would otherwise be expected to fare better based on their classification (e.g., GS = 6: UALCAN (uab.edu) (second page) (Figure 1)). With over 100 identified proteins that are direct interactors of TLK1 [38], attempting to pinpoint those that may directly impact poor survival was a challenge.

Due to its potential signaling function, we investigated the role of a kinase relay mechanism involving MK5, and we obtained considerable evidence that the TLK1>MK5 axis is an important node/regulator of cell motility, cytoskeletal rearrangements, invasion through ECM, and ultimately metastatic spread [15,21]. Indeed, one of the most convincing aspects of this is the ability of overexpressed TLK1 (OE, a mechanism commonly exploited by PCa cells upon ADT [15,21]) to induce the relocalization of GFP-MK5 from nuclei to the actin cytoskeleton (Figure 2), orchestrating pro-motility lamellipodial rearrangements [15]. This is attributable to its phosphorylation at 3 sites—S160, S354, and S386—and alterations in the subcellular distribution of MK5 were implicated previously [26]. We had mechanistically studied in detail these EMT plasticity [39,40] and motility/invasion changes [15,21], as we proposed they may relate to metastatic behavior. Still, they had not addressed sufficiently the therapeutic and diagnostic implications, which we do in this work. The effect of inactivating TLK1 on phosphorylation of MK5-S354 is visually depicted in Appendix A, which indicates a direct dependence on it.

### 2.2. Reduction of Metastatic Spread with Inhibitors of TLK or MK5

Experimental metastasis was studied using an established model [41] using AR^−^ PC3-Luc cells. A total of 5 × 10^5^ PC3-Luc cells, which express both TLK and MK5, were injected into the lateral veins of the mice. In our initial work [15] and repeated here, we have found consistently that after 5 weeks, this results in 10–40 clearly visible lung nodules (Figure 3). Evidence of metastases was found primarily in the lungs, with a few tumors found at the base of the tail (probably at the site of IV injection) and a few in the liver. Bone metastases (poorly visible in some limbs and occasional mandibula, as in examples at the bottom right) were better studied in the following tibia inoculation model. We have administered J54 (TLK inhibitor) or GLPG (MK5 inhibitor) IP every third day from the day after inoculation of cells. Mice were first imaged with the IVIS after 1 week under isoflurane sedation and sacrificed 1 month later. The Kruskal–Wallis test in GraphPad Prism was used for statistical analyses of differences in tumor nodule burden. These experiments support the hypothesis that the TLK1>MK5-dependent invasion/motility is a key to metastatic spread (H(2) = 17.91, *p* = 0.0001; Dunn’s post hoc of 0.0012 and 0.0005 between control vs. GLPG and control vs. J54, respectively) and can be targeted to hinder metastasis with low toxicity using a repurposed ‘MK5 inhibitor’ that has already passed FDA/Phase I-II approval, although found to be ineffective as a treatment for arthritis [42,43] or J54. Note that in this experiment shown in Figure 3, GLPG did not reduce the tumor growth (see size in 4th lung set in C) but reduced the number of lung (and bones) metastases in a dose-dependent fashion. In contrast, J54 (TLK1i) also reduced the size of tumor nodules (Figure 3E,F) and cell proliferation previously reported by IHC by the number of Ki67-positive cells [14,15]. It is likely that inhibition of TLK1 with J54 also affects the Nek1>YAP tumor-promotion pathway [34,44]. We tried to verify that the inhibitors (mainly for J54) had the biochemically intended effect by monitoring the phosphorylation of pMK5-S354. Note that J54 inhibits TLK1 activity, not expression, and similarly, GLPG inhibits MK5 activity and not its expression. However, as shown in Appendix A, pMK5-S354 (a direct substrate of TLK1) is strongly reduced in selected lung tumors of mice treated with J54. In addition, J54 increased somewhat the level of TLK1 by locking it in a conformation-stabilized inactive complex (as we previously reported in [14]). There are no similar reliable assays for readout of Mk5 activity or its inhibition.

### 2.3. Role of TLK1>MK5 in Cancer Dissemination in a Mouse Model of Spontaneous PCa Progression

As stated above, the TRAMP mouse presents a useful and well-established model of age-related PCa progression all the way to metastases of the liver and lungs. Two sets of experiments were implemented to study the involvement of these two sequentially acting kinases during late-stage cancer progression in the TRAMP model: (1) We tested the effect of inducing the expression of TLK1B with an ARSI (ENZ) to see if there was an increase in livers and lungs metastases and, conversely, if inhibiting the activity of TLK1/1B directly resulted in reduced distal organs metastases. (2) We tested a genetic model where we crossed the TRAMP mice with the MK5-KO strain. The results shown in Figure 4, displayed as a qualitative chart on a 0–4 scale for the presence of tumor burden (averaged between the number and size of nodules), showed that whereas cancer of the prostate occurred in all study groups, the distal metastases were significantly different. For the liver, the Kruskal–Wallis test showed a significant difference in tumor burden (H(4) = 32.63, *p* < 0.0001) among the groups. Post hoc comparisons using Dunn’s test indicated that the differences between the TRAMP control and TRAMP-MK5KO (CI = [19.35, 20.15]), as well as between TRAMP control and TRAMP ENZ+ J54 (CI = [15.82, 16.68]), were significant (0.0041 and 0.0366, respectively). For the lungs, the Kruskal–Wallis test revealed a significant difference in tumor burden (H(4) = 33.94, *p* < 0.0001). Post hoc comparisons using Dunn’s test indicated that the differences between the TRAMP control and TRAMP-MK5KO (CI = [22.26, 23.12]) and between TRAMP control and TRAMP ENZ+J54 (CI = [19.68, 20.58]) were significant (0.0002 and 0.0018, respectively). A large effect size (h2 = 0.8 for both results) suggests that the group differences account for a substantial portion of the total variability, indicating that the differences between the groups are highly meaningful and significantly impact tumor burden. At the same time, the combination ENZ+J54 significantly reduced tumor burden in what we have described as anti-ARSI addiction on the TLK1>MK5 pathway. Genetically, the TRAMP.MK5-KO mice, for the most part, presented rarely with distal tumors.

### 2.4. Evaluation of PCa Progression in TRAMP via IHC for pMK5 Ab

One of the identified phosphorylation sites of TLK at MK5 is the serine residue 354 in the nuclear localization signal of MK5 (cf. Figure 2). The pattern of pMK5 staining was monitored in TRAMP mice at different ages (Figure 5), from 8 w (benign hyperplasia and PIN) to 12 w (PIN and well-differentiated PRAD) to 20 w (invasive and disorganized PRAD). While cytoplasmic/nuclear compartment ratios are small in cancer cells, and the stain intensity does not change much from benign to malignant cells, it would appear that the cytoplasmic staining intensity of p354-MK5 increases in these age-related lesions.

### 2.5. Evaluation of PCa TMA with pMK5 Ab

If the TLK-MK5 pathway is a key driver of invasion and metastatic spread, increased levels of pMK5 should be seen in samples with higher GS and/or the presence of N, M, as we reported on the initial evaluation [21]. Also, there should be a decrease in the nuclear-to-cytoplasm ratio in the N^+^ patients (compare Figure 2 and IHC in Figure 6), akin to the observed GFP-MK5 relocalization upon TLK1 OE [21]. In this case, pMK5 may yield valuable prognostic information, especially in post-treatment cases (e.g., neoadjuvant ADT before prostatectomy or XRT) where the GS can be misrepresentative of actual pathology due to treatment-induced morphologic changes [45].

### 2.6. Direct Bone Engraftment

The tibia inoculation system was used to study the efficiency of engraftment. Metastasis is a complex process that is subdivided into, at minimum, three key phases: (1) detachment/shedding from the primary site requires local invasion and intravasation into the bloodstream; (2) survival in the bloodstream shear force, adhesion to a capillary wall, and commencing extravasation at the distal site; and (3) colonization and proliferation at new metastatic sites. If any of these three processes fail, cancer cells cannot survive, resulting in the eventual elimination from circulation, regardless of their entity in liquid biopsies [46,47]. Our work so far, and particularly the results from the lungs dissemination after tail-vein injection (where step 1 is bypassed), suggest that the TLK1>MK5 pathway and its inhibition by GLPG or J54 more likely affect one or more processes, either inhibiting survival, preventing extravasation, or affecting engrafting and proliferation in the alveolar tissue. The tibia engrafting model is a more direct test for the ability of the injected cancer cells to reshape the local ECM and colonize the metastatic niche.

For the tibia engraftment experiment, four mice per group were inoculated with 10^6^ PC3-Luc cells that were pre-treated overnight with 10 µM GLPG or J54 before injection in both legs, and just four days later, imaging with the IVIS was carried out. We had previously determined that neither treatment affects the viability of the cells and most of their metabolic features. However, GLPG was found to reduce their motility and invasion through Matrigel plugs [15]. The result was that clearly, both treatments were effective at strongly reducing active engraftment, as >50% of the mice showed no signal in either bone side. This clearly indicated that the TLK1>MK5 is very important for establishing engraftment at the distal site, at least in the case of bone (a primary metastasis target for PCa). However, there was another subtle difference. It appeared that where engraftment did occur in the mice inoculated with cells pre-treated with J54, the tumors’ luminescence indicated that the cells’ growth was unimpeded. However, in the cells pre-treated with GLPG, in the single mouse where engrafting was evident, the growth of the tumor was also impaired and therefore the signal was weaker (see ROI signals in Figure 7). This feature, while admittedly eight tibias per group are insufficient for solid statistics, was actually the opposite of what we had concluded for the lung metastases experiments determined via the tail-vein injection, where the mice treated with J54 showed reduced tumor sizes compared to those treated with GLPG. While we are clearly dealing with different models/methods of tumor engagements, our interpretation of the most likely difference between these two conditions is attributable to the respective TME, where the alveolar (lung tissue) vs. trabecular (bone) spaces clearly have different properties. Also, the stiffness of the TME might be differently affected by treatment, as it is well known that cancer cells can reshape their own TME during seeding. An alternative explanation could be that the number of cells arriving at the distal site and extravasating (a motility/invasion feature) in either inoculation condition is very different at the onset and thus affects early seeding/colonization. The fact that the host is immunocompromised and the cells were inoculated pre-treated (rather than via systemic treatment of the mice) implicates that the effect observed on the prevention of tumor engraftment is ‘cell autonomous’ and should be potentially ‘translatable’ to other PCa cell types with other genetic/epigenetic backgrounds and phenotypes.

## *3.* Discussion

### 3.1. Choice of GLPG vs. J54 in Clinical Translation

Regarding the experimental metastases results, it was noticeable that GLPG did not reduce the size of the tumor nodules in the lungs (hematogenous route) compared to untreated animals but clearly reduced the number of lung metastases. This would argue against the direct involvement of the mTORC1>TLK1B pathway [8,14], as this would likely also affect proliferation. In contrast, the TLK1 inhibitor J54 also reduced the size of the tumors, visibly so at lung dissection and also histologically for micrometastases that were generally negative for Ki-67 staining [15]. This suggests that, in vivo, J54 may also modulate the proliferation of cancer cells, a property that we attributed earlier to the TLK1>NEK1>YAP1 axis [34] that largely affects the contact inhibition features (mechano-transduction) of the cells [48]. In fact, J54 alone strongly inhibited the growth rate of VCaP-Luc subcutaneous tumors [44]. While the processes orchestrating tumor growth and metastasis are distinct, they are also inextricably linked [3,49,50,51]. Target validation via monitoring the pNek1-T141>pYAP-Y407 axis was also shown via WB analysis of the excised tumors [44], and interestingly, the expression of PD-L1 (a target of YAP [52]) was likewise suppressed with J54, with important implications for tumor suppression when using immunocompetent mice or men in a future clinical trial. In addition, as TLK1 plays important functions in the DDR and DNA repair [9], its inhibition in replication-stressed cancer cells could promote genomic instability and the generation of neoantigens, upon which a more effective immune response may ensue to curtail the rapid growth of tumors [53]. However, GLPG is further along in clinical testing (completed Phase I-II for arthritis indication [43]). We are aware that the conclusions were based on relatively small sample sizes and that despite the support of convincing statistical analyses, they will require the verification of multi-center collaboration studies before applying for clinical trials.

Clinical Implications for PCa Patients Selection: It appears that employing an approach to inhibit TLK1-MK5 in localized prostate cancer to prevent metastasis presents a challenging opportunity for clinical adoption, as the timing and duration of such therapy do not seem to provide a clear clinical window. However, with careful monitoring, we propose that the ideal target patients are those with biochemical signs of recurrent disease (an increase in PSA) but without radiologic evidence of widespread metastases, such as indicated by a PSMA-PET scan.

### 3.2. Specificity vs. General Toxicity Considerations

TLK1 has ~150 protein targets [38], while the list for MK5 is more limited and possibly incomplete [54]. Despite such expected biomic complexity, TLK1-KO and MK5-KO mice are viable and fertile. This lack of ‘essentiality’ makes them an excellent target for intervention upon their metastatic function, with a lesser concern for other important issues for viability and well-being. However, we expect and acknowledge their potential existence during future clinical development.

A discussion of specificity cannot avoid consideration of the implied mechanisms of action of the two kinases, separately and together. As far as TLK1 and its proposed role in the cancer progression leading to metastasis, several reviews have already been published [8,20], while several reviews of the possible activities of MK5 in cancer progression can be found in [25,32,55,56,57]. However, its connection with the specific activity/role played by MK5 is less clear. TLK1-mediated phosphorylation of MK5 in three novel residues (S160, S354, and S386) may promote the shuttling of MK5 out of the nucleus and within the cytoplasm, where it exerts its pro-metastatic function by modulating the phosphorylation of focal adhesion proteins (pFAK Y861 and pPaxillin Y118) and HSP27 (pHSP27-S82) [58,59,60] that enhances actin filamentation and reorganizations in the leading edge of the cells. Furthermore, TLK1>MK5-mediated ERK3 stabilization stimulates several MMPs (MMP2 and MMP9) expressions, enhancing the invasive capacities of the PCa cells and their TME interaction to orchestrate functional PCa metastasis [15,21]. Of the three identified phosphorylation sites on MK5, we only investigated in more detail the effect of the S354A replacement in MK5^−/−^ MEFs, where we found that the cells failed to restore their motility compared to that of wild-type WT-MK5 rescued MK5^−/−^ MEF cells, just as much as the M5-kinase-dead K51E [21].

## 4. Conclusions

We have used two main models to generate more evidence about the potential of targeting the TLK1>MK5 axis in order to suppress metastatic spread. One is the highly invasive and metastatic cell line PC3, which is AR-negative and refractory to ARSI, thus beyond the current treatment response capacity of the current standard of care. A minor fraction of these cells also represents a population of cancer stem cells with remarkably high replication capacity [61]. The other model was a GEMM specifically generated with a cross of the TRAMP mouse on MK5-KO background to monitor the contribution of MK5 to spontaneous, age-related metastases. Many researchers do not like the TRAMP model, as it is based on T-Ag transformation and not typical human oncogenes, although it is far simpler than those that require multiple crosses with CRE hosts. Despite the well-documented advantages of the TRAMP model, certain limitations persist. The probasin promoter is androgen-regulated, but tumors arising from SV40 antigen transgene expression are androgen-independent. Early castration or hormone ablation may misleadingly suggest a refractory therapeutic response period, not due to treatment efficacy but rather to suppressed transgene expression. Additionally, after castration, tumors develop rapidly and tend to be more poorly differentiated, pathologically aggressive, and highly metastatic. In patients, ADT regimens have shown improvement in PCa management and prolonged survival (up to 6 years) before progression to more aggressive metastatic disease. However, despite the limitations of using this model for representation of human PCa, it is quite adequate for the study of late-age metastases, and it is also a model that finally converts with high frequency from PRAD to CRPC/NEPC [16]. Our preferred alternative model is the PCa-targeted PTEN-KO. The PTEN-KO has been described in detail in [62]. The prostate tumors that develop in mice with homozygous and +/− PTEN deletion are AR+ adenocarcinomas that, in most castrated mice, progress to mCRPC. PIN lesions develop at 6 weeks in 100% of homozygous mice and progress to invasive carcinomas by 9 weeks. A key feature of these lesions is the activation of AKT, which can be used as a phenotypic marker. An obvious expectation is that this will result in mTOR activation and hence overexpression of TLK1B and its consequent activation of Nek1>YAP and MK5 pathways, leading the way to mCRPC progression if the mice are castrated. This model mirrors the majority of advanced human PCa (15–20% in primary cancers and increasing to 40–60% in more aggressive stages like castration-resistant and metastatic disease [63,64,65,66]), and hence, if treatment with J54 prevents mCRPC dissemination in castrated animals, it would provide a strong pre-clinical test for its use in combination with ARSI.

In conclusion, while we are unable to make wide claims about the direct applicability of our studies to help in the management of mCRPC, we did provide some solid evidence for the role of the TLK1>MK5 axis in this progression, corroborated by a correlative PCa-TMA analysis of patients who presented (or not) lymph node involvement more frequently when the pMK5 stain was strong and cytoplasmic. This fact alone could be used by pathologists and oncologists for making more informed, aggressive decisions on the next course of action following the initial surgical biopsy analysis, when a sentinel node dissection, when performed, may miss the detection of cancer cells’ spread [67,68].

## 5. Materials and Methods

An RPMI-1640 medium was purchased from Thermofisher (Houston, TX, USA). CPT was purchased from Millipore-Sigma (St Louis, MI, USA-P4394). The J54 was synthesized by our group as described in the STAR Methods of [14]. However, we are aware that it is now sold by Probechem (TLK1 inhibitor J54|TLK1 inhibitor|Probechem Biochemicals, Shanghai, China). Hek293 cells expressing GFP-MK5 and TLK1 were previously described [15,21].

### 5.1. Cell Viability Assay and Treatments

Cell viability was evaluated using the MTT assay, which measures the reduction of 3-(4, 5-dimethylthiazol-2-yl)-2, 5-diphenyl tetrazolium bromide (MTT) by mitochondrial enzymes in live cells. We seeded 20,000 PC-3 and C4-2B cells into 100 μL of the medium in 96-well plates and allowed them to adhere for 24 h. Then, we replaced the medium with a fresh medium containing various cisplatin and/or J54 concentrations and incubated them for an additional 24 h. Finally, we added the MTT reagent to each well and incubated it for 35 min before measuring the absorbance intensity at 490 nm using a spectrophotometer. PC3 cells were treated with 10 µM of GLPG 0259 or J54 (Medkoo Biosciences, Inc., Morrisville, NC, USA, cat# 561481; CAT#: 556219) for 48 h in T-75 flasks until confluency. DMSO-treated cells were considered as vehicle control (VC). After the treatment, cells were harvested for injection after resuspension in a minimal volume of their conditioned medium.

### 5.2. Animal Studies

All animals used in this study received humane care based on the recommendations set by the American Veterinary Medical Association, and the Institutional Animal Care and Use Committee of the LSU Health Sciences Center at Shreveport approved all the test protocols. Immune-deficient NOD SCID mice (Charles River, Skokie IL, USA) were used in this research to host human PCa PC-3 tumors; 0.5–1 × 10^6^ human PCa PC-3-Luc cells were suspended in condition medium (with drug where indicated) and injected in either the tail vein or grafted in the tibia. For treatment, J54 or GLPG dissolved in 200 sterile saline with 10% Polysorbate-80—PS-80) was given bi-weekly IP. In vivo imaging was done using an IVIS-spectrum/CT machine (Perkin Elmer, Shelton, CT, USA).

### 5.3. Immunohistochemistry and Fluorescence Imaging

These were standard, and as previously described, for mouse IHC staining, tissues were harvested, formalin-fixed, processed, and paraffin-embedded. Prostate TMA was prepared. For antigen retrieval, tumor tissues were serially sliced into 5-µm-thin sections, deparaffinized in xylene, rehydrated with lower ethanol concentrations, and boiled in an unmasking solution (10 mm sodium citrate + 1 mm EDTA) for 15 min. A 3% hydrogen peroxide was used to quench the cellular peroxidase. The primary antibody against pMK5 Ser354 (1:200 dilutions) was then incubated for an entire night at 4 °C after tissue sections had been blocked for one hour in 3% BSA. After three rounds of washing, the sections were incubated for one hour at room temperature in secondary antibody (Vector Laboratories, Burlingame, CA, USA, cat# PK-6200). The slices were cleaned and then incubated for 30 min with Vectastain ABC reagent (Vector Laboratories, Burlingame, CA, USA, cat# PK-6200). After another wash, the sections were incubated for two minutes in DAB substrates (Vector Laboratories, Burlingame, CA, USA, cat# SK-4105). Thereafter, the slides were cleaned, dehydrated, coverslipped, and counterstained with hematoxylin. Quantification and imaging were carried out. The anti-pMK5-S354 is from Thermofisher CAT:A5-105676.

### 5.4. Statistical Analysis

GraphPad Prism 9 (GraphPad Software Inc., San Diego, CA, USA) was used for statistical analysis. The Kruskal–Wallis test was conducted to compare multiple groups, followed by Dunn’s post hoc analysis; *p*-values < 0.05 were considered significant. For the TLK1>MK5-dependent invasion/motility in metastatic spread, the Kruskal–Wallis test demonstrated a significant difference (H(2) = 17.91, *p* = 0.0001) in tumor burden among the groups. Dunn’s post hoc tests of 0.0012 and 0.0005 between control vs. GLPG (CI = [13.35, 14.55]) and control vs. J54 (CI = [14.30, 15.40]), respectively, indicate a significant difference in tumor nodules resulting from the drug treatment. An effect size of h2 = 0.6 suggests meaningful differences among groups that significantly influence the tumor nodules. Outliers were identified before analysis using the interquartile range (IQR) rule (1.5*IQR). However, the analysis was conducted both with and without the outlier. The results from the Kruskal–Wallis and Dunn’s tests are robust against outliers; the outliers did not significantly change the results.

## Figures and Tables

**Figure 1 cancers-17-01187-f001:**
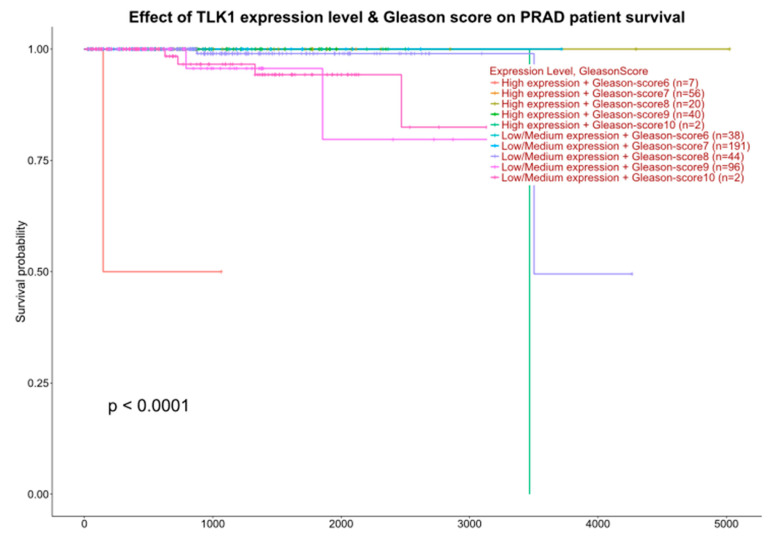
Expression analyses of TLK1 relation to outcome.

**Figure 2 cancers-17-01187-f002:**
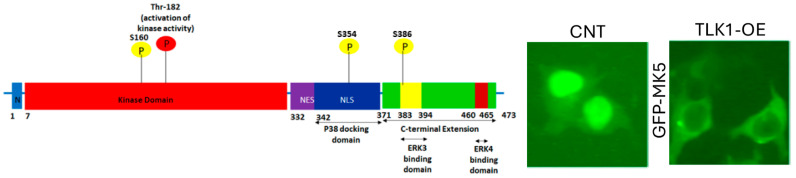
(**Left**): Mapping of the 3 novel phosphorylation sites (shown as yellow circles with corresponding residue number) on the MK5 functional domains. (**Right**): Subcellular localization of ectopically expressed GFP-tagged MK5 in Hek293 cells. Overexpression of TLK1 causes translocation of MK5 from the nucleus to the cytoplasm.

**Figure 3 cancers-17-01187-f003:**
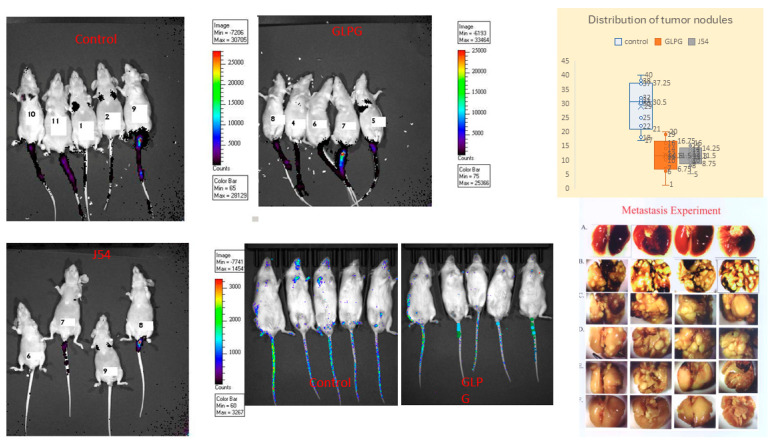
Experimental metastasis via injection of PC-3 cells in the lateral tail vein. The metastases in the lungs were imaged by IVIS after 1 week (note reduced ‘take’ in treated groups) or scored 5 weeks after inoculation (right). (**A**), not inoculated control lungs. (**B**), inoculated, vehicle control. (**C**,**D**), GLPG at 2 or 5 mg/kg. (**E**,**F**), J54 at 2 or 10 mg/kg (treatment twice/w). Lung nodules are quantified on Top (*p* < 0.001 compared to untreated group). The boxplot is for groups (**B**,**D**,**F**).

**Figure 4 cancers-17-01187-f004:**
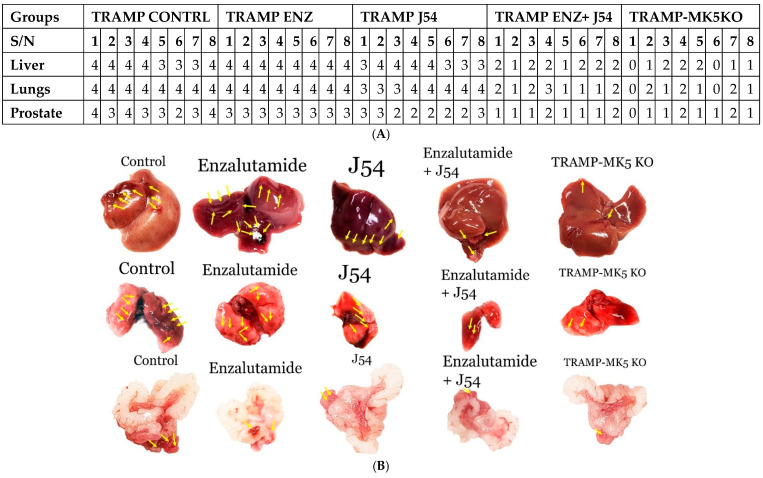
TLK1Bi, TLK1/MK5KO reduced distal organ metastasis in TRAMP mouse model. Mice were randomly distributed into groups of 8 and were bred until 13 weeks. Thereafter, oral treatment with enzalutamide (ENZ) and/or J54 was given twice per week for 13 weeks (control and TRAMP-MK5 KOs were given PBS orally). Subsequently, mice were euthanized and the respective organs (lungs, liver, and prostate) removed. (**A**) Table showing the qualitative scores of tumor burden per mouse. 0—No burden, 1—very low burden, 2—low burden, 3—moderate burden, 4—high burden. The qualitative scores of tumor burdens show the presence of prostate tumor in all groups. However, distal metastases to the lungs and liver were significantly reduced in the TRAMP-MK5-KO and TRAMP-ENZ+ J54 groups (*p* < 0.0001). (**B**) Representative images of liver, lungs, and prostate image of vehicle control, TRAMP treated with J54 (10 mg/kg) and/or enzalutamide (10 mg/kg) twice a week, and TRAMP-MK5 KO at age 26 weeks. We did not observe differences between homozygous MK5-KO vs. heterozygous.

**Figure 5 cancers-17-01187-f005:**
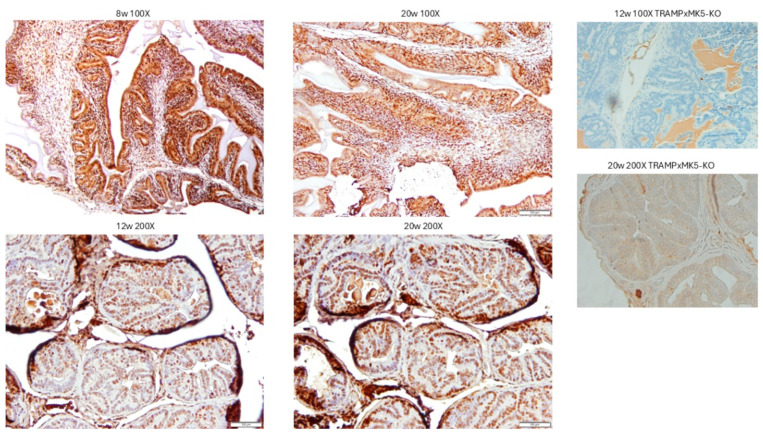
IHC staining revealed elevated pMK5 Ser354 level in PCa progression of TRAMP mice. Representative images of TRAMP prostates; thin sections from mice sacrificed at different ages and stained by IHC for pMK5 are shown.

**Figure 6 cancers-17-01187-f006:**
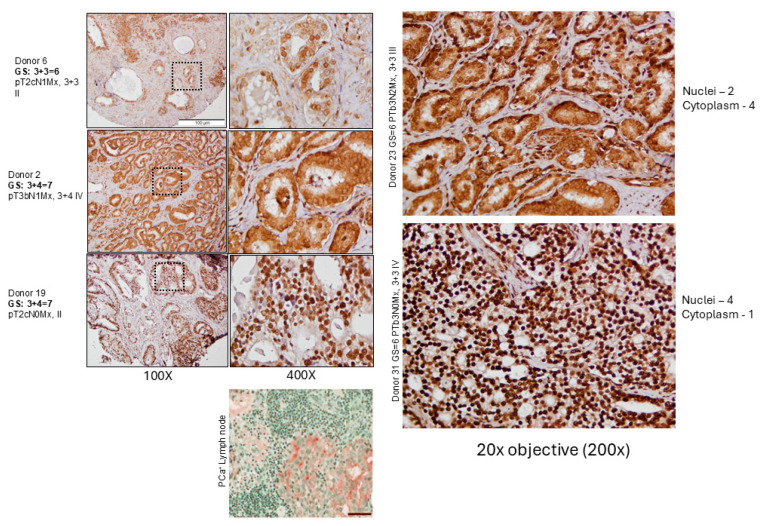
TMA/IHC staining revealed elevated pMK5-S354 (cytoplasmic) staining in patients with high metastatic penetrance. TMA samples of similar GS differ in intensity of pMK5 staining and nuclear/cytoplasmic ratio that correlates with nodal metastasis. MK5 is generally localized to the nuclei, but shuttles to the cytoplasm upon activation, where it is believed to promote reorganization of the cytoskeletal network. We selected a few, most representative images from our institutional TMA stained with pMK5 (Thermofisher A5-105676). One can see that cytoplasmic staining matches more closely with presence of lymph metastasis involvement (N+); an example of PCa^+^ lymph node shows largely cytoplasmic staining in the nest. Stain intensity on 0–4 scale. Note the absence of stain in the MK5-KO PCa tissue. Scale Bar 50 µm.

**Figure 7 cancers-17-01187-f007:**
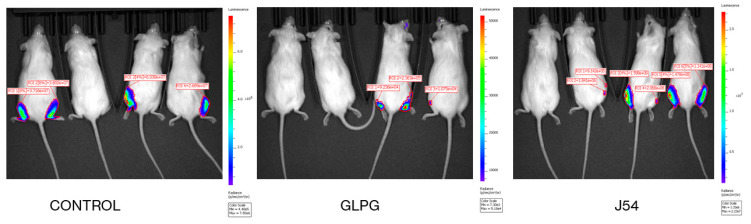
Tibia engraftment of PC3-Luc cells. PC3-Luc cells were pre-treated with GLPG or J54 (10 µM) for 12 h. After resuspension at 50,000 cells/µL in condition medium, they were inoculated with a Hamilton syringe in each tibia under isoflurane anesthesia. Four days after recovery, the mice were imaged with the IVIS and the signals from the tumors in each leg were quantitated (ROI).

## Data Availability

A description of all data and materials can be found in the referenced article. No additional data have been withheld from the public.

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
