# Peer review of "Targeting the TLK1-MK5 Axis Suppresses Prostate Cancer Metastasis"

_cancers, 2025, doi:10.3390/cancers17071187_

Round 1
Reviewer 1 Report
Comments and Suggestions for Authors
An interesting follow-up on targeting the TLK1-MK5 axis; however, the study lacks sufficient supporting evidence in experimental design and data interpretation.
- Materials and Methods
This section is overly simplified and lacks essential details necessary for reliable validation in future studies. For example, it does not include information on statistical analysis.
3.1. Interrogation of Expression Reports
This section reads more like an introduction and rationale for targeting the TLK1 gene rather than a presentation of experimental results. Additionally, Figure 1 is poorly explained—it is unclear whether the data is directly sourced from a website or further analyzed by the authors. This section should be removed from the Results and considered for integration into the Introduction instead.
3.2. Reduction of Metastatic Spread with Inhibitors of TLK or MK5
"In our initial work [31] and repeated here, we have consistently found that after five weeks, this results in 10–40 clearly visible lung nodules (Fig. 3)."
- Figure 3 is poorly organized. The quality of Figures 3A–F from the metastasis experiments is low (it appears to be copied from a poster or slides).
- There is a lack of visible quantification and statistical graph comparisons of the number of nodules between different groups.
- The grouping of mice in panels is unclear, making the results confusing. Detailed explanations are needed.
- There is no validation of TLK or MK5 level changes following the administration of J54 or MK5 inhibitors.
- Evidence of successful inhibition of TLK or MK5 is required.
3.3. Role of TLK1>MK5 in Cancer Dissemination in a Mouse Model of Spontaneous PCa Progression
"The results, shown in Figure 4, are displayed as a qualitative chart on a 0–4 scale:"
"0 – No burden, 1 – Very low burden, 2 – Low burden, 3 – Moderate burden, 4 – High burden."
- The criteria distinguishing each scale level are unclear. Examples illustrating metastasis burden levels from 1 to 4 are needed.
- No statistical analyses were conducted between different groups.
- This section lacks a clear conclusion or interpretation based on the findings.
3.4. Evaluation of PCa Progression in TRAMP via IHC for pMK5 Ab
3.5. Evaluation of PCa TMA with pMK5 Ab
"One of the identified phosphorylation sites of TLK at MK5 is the serine residue 354/220 in the nuclear localization signal of MK5 (cf. Fig. 2)."
- The order of Figure 2 should be revised for better coherence.
- The IHC staining images in Figure 5 appear to be duplicated in Figure 6. This should be corrected.
Author Response
Reviewer 1
Comments and Suggestions for Authors
An interesting follow-up on targeting the TLK1-MK5 axis; however, the study lacks sufficient supporting evidence in experimental design and data interpretation.
- Materials and Methods
This section is overly simplified and lacks essential details necessary for reliable validation in future studies. For example, it does not include information on statistical analysis.
We have expanded some of the Methods and explained the statistics.
3.1. Interrogation of Expression Reports
This section reads more like an introduction and rationale for targeting the TLK1 gene rather than a presentation of experimental results. Additionally, Figure 1 is poorly explained—it is unclear whether the data is directly sourced from a website or further analyzed by the authors. This section should be removed from the Results and considered for integration into the Introduction instead.
It was taken directly from the UALcan analysis (with a link provided). However, the figure was left as is as it would have been difficult to refer to it otherwise.
3.2. Reduction of Metastatic Spread with Inhibitors of TLK or MK5
"In our initial work [31] and repeated here, we have consistently found that after five weeks, this results in 10–40 clearly visible lung nodules (Fig. 3)."
- Figure 3 is poorly organized. The quality of Figures 3A–F from the metastasis experiments is low (it appears to be copied from a poster or slides).
The figure was replaced with a higher quality one.
- There is a lack of visible quantification and statistical graph comparisons of the number of nodules between different groups.
The boxplot has the ‘real’ number of visible mets for each 10-mice groups B, D, F
- The grouping of mice in panels is unclear, making the results confusing. Detailed explanations are needed.
We have better detailed the groups analyzed in boxblot.
- There is no validation of TLK or MK5 level changes following the administration of J54 or MK5 inhibitors.
Added a figure in SI. J54 inhibits TLK1 activity – not expression, and similarly, GLPG inhibits MK5 activity and not expression. However, as shown in SI1, pMK5-S354 (a direct substrate of TLK1) is strongly reduced in selected lung tumors of mice treated with J54. In addition, J54 actually increases somewhat the level of TLK1 by locking it in a conformation-stabilized inactive complex (previously reported in the iScience paper (https://doi.org/10.1016/j.isci.2020.101474). In addition, we have added an illustration of the effect of inhibiting TLK1 on pMK5-S354 in Fig.SI1.
- Evidence of successful inhibition of TLK or MK5 is required.
See answer above.
3.3. Role of TLK1>MK5 in Cancer Dissemination in a Mouse Model of Spontaneous PCa Progression
"The results, shown in Figure 4, are displayed as a qualitative chart on a 0–4 scale:"
"0 – No burden, 1 – Very low burden, 2 – Low burden, 3 – Moderate burden, 4 – High burden."
- The criteria distinguishing each scale level are unclear. Examples illustrating metastasis burden levels from 1 to 4 are needed.
- No statistical analyses were conducted between different groups.
These are now shown as test for non-parametric analysis ( Kruskal-Wallis test ) with Dunn’s Post-hoc analysis.
- This section lacks a clear conclusion or interpretation based on the findings.
The interpretation is now better explained
3.4. Evaluation of PCa Progression in TRAMP via IHC for pMK5 Ab
3.5. Evaluation of PCa TMA with pMK5 Ab
"One of the identified phosphorylation sites of TLK at MK5 is the serine residue 354/220 in the nuclear localization signal of MK5 (cf. Fig. 2)."
- The order of Figure 2 should be revised for better coherence.
switched
- The IHC staining images in Figure 5 appear to be duplicated in Figure 6. This should be corrected.
Accidental duplication corrected. Thank you for accepting its intended meaning nonetheless.
Reviewer 2 Report
Comments and Suggestions for Authors
This article investigates the role of the TLK1-MK5 signaling pathway in prostate cancer metastasis and explores the potential to inhibit prostate cancer metastasis by targeting this pathway. Using pharmacological and genetic approaches, combined with mouse models, the authors found that the TLK1-MK5 pathway plays an important role in the motility, invasiveness, and metastatic potential of prostate cancer cells. The study also showed that the existing TLK1 inhibitor J54 and MK5 inhibitor GLPG0259 have low toxicity and can significantly reduce the number and size of metastatic lesions. This study provides valuable insights into the development of new therapeutic strategies, especially in the treatment of metastatic prostate cancer that is refractory to anti-androgen therapy. However, this work still needs to address some important issues. It is hoped that by addressing these issues, the overall quality of the paper can be improved.
General Comments:
- The authors employ both pharmacological and genetic approaches, using both in vitro and in vivo models. The use of the TRAMP mouse model is appropriate and provides strong evidence for the relevance of TLK1 and MK5 in metastasis. While the study design is solid, some additional clarification is needed regarding the dosing of GLPG0259 and J54 in animal experiments, as well as the rationale for selecting specific metastasis models (e.g., tibia vs. tail-vein injection).
- Data Analysis: The statistical methods used (e.g., chi-squared tests and IVIS imaging) are appropriate for the dataset. However, further clarification on effect sizes and confidence intervals would strengthen the overall analysis. Additionally, the authors should discuss how outliers were handled in the data.
- Novelty and Impact: The manuscript explores the role of the TLK1-MK5 signaling axis in prostate cancer (PCa) metastasis, an important area of research. The study demonstrates the potential of targeting this axis as a therapeutic strategy, particularly in androgen receptor-independent metastatic prostate cancer. The identification of two inhibitors, GLPG0259 and J54, with pre-existing clinical safety data provides promising clinical applicability. However, the novelty of the study might be enhanced by further elaboration on how this axis interacts with other established pathways in PCa metastasis.
Specific Comments:
- Abstract and Introduction:
- The abstract provides a clear summary of the findings. However, the introduction could benefit from a more concise presentation of previous research that identifies TLK1 and MK5 as key players in PCa metastasis. This would help contextualize the study more clearly.
- Results section:
2.1 The data supporting the role of TLK1 and MK5 in metastasis is compelling. However, some figures, particularly Figure 4, could benefit from clearer labeling and more consistent comparisons between experimental groups. The figure captions should be more detailed to help the reader understand the key findings.
2.2 The experiment involving J54 and GLPG should be described in more detail, particularly regarding the doses and treatment schedules. It would also be helpful to include additional data on toxicity or adverse effects, even if they were minimal.
- Discussion:
3.1 The discussion section briefly summarizes the key findings, but it could be expanded to include a more detailed interpretation of how these results fit within the broader context of cancer metastasis research. Specifically, the authors should explore the potential implications of targeting the TLK1-MK5 axis in combination with other therapies.
3.2 The experimental validation of the molecular mechanisms underlying TLK1-MK5-mediated metastasis is currently lacking. The authors should propose specific in vitro and in vivo experiments to explore the exact molecular interactions that promote metastatic behavior in PCa.
3.3 The manuscript would benefit from a more thorough discussion of the limitations of the study. For example, while the use of the TRAMP mouse model is valuable, it would be helpful to discuss the generalizability of these findings to human cancers and other animal models.
Comments on the Quality of English Language
English should be improved.
Author Response
Reviewer 2
Comments and Suggestions for Authors
This article investigates the role of the TLK1-MK5 signaling pathway in prostate cancer metastasis and explores the potential to inhibit prostate cancer metastasis by targeting this pathway. Using pharmacological and genetic approaches, combined with mouse models, the authors found that the TLK1-MK5 pathway plays an important role in the motility, invasiveness, and metastatic potential of prostate cancer cells. The study also showed that the existing TLK1 inhibitor J54 and MK5 inhibitor GLPG0259 have low toxicity and can significantly reduce the number and size of metastatic lesions. This study provides valuable insights into the development of new therapeutic strategies, especially in the treatment of metastatic prostate cancer that is refractory to anti-androgen therapy. However, this work still needs to address some important issues. It is hoped that by addressing these issues, the overall quality of the paper can be improved.
General Comments:
- The authors employ both pharmacological and genetic approaches, using both in vitro and in vivo models. The use of the TRAMP mouse model is appropriate and provides strong evidence for the relevance of TLK1 and MK5 in metastasis. While the study design is solid, some additional clarification is needed regarding the dosing of GLPG0259 and J54 in animal experiments, as well as the rationale for selecting specific metastasis models (e.g., tibia vs. tail-vein injection).
Dosing was established in previous work and reported again in the legend. Both tail-vein and tibia inoculation were selected for their rapid ‘take’ at the most proximal target site, unlike the orthoptic (intraprostatic) spontaneous metastasis model that can take months to develop. For spontaneous metastasis we used instead the GEMM with TRAMP/MK5-KO.
- Data Analysis: The statistical methods used (e.g., chi-squared tests and IVIS imaging) are appropriate for the dataset. However, further clarification on effect sizes and confidence intervals would strengthen the overall analysis. Additionally, the authors should discuss how outliers were handled in the data.
This is now better detailed.
- Novelty and Impact: The manuscript explores the role of the TLK1-MK5 signaling axis in prostate cancer (PCa) metastasis, an important area of research. The study demonstrates the potential of targeting this axis as a therapeutic strategy, particularly in androgen receptor-independent metastatic prostate cancer. The identification of two inhibitors, GLPG0259 and J54, with pre-existing clinical safety data provides promising clinical applicability. However, the novelty of the study might be enhanced by further elaboration on how this axis interacts with other established pathways in PCa metastasis.
I have added a discussion of the PTEN and TMPRSS2-ERG
Specific Comments:
- Abstract and Introduction:
- The abstract provides a clear summary of the findings. However, the introduction could benefit from a more concise presentation of previous research that identifies TLK1 and MK5 as key players in PCa metastasis. This would help contextualize the study more clearly.
We have worked on this.
- Results section:
2.1 The data supporting the role of TLK1 and MK5 in metastasis is compelling. However, some figures, particularly Figure 4, could benefit from clearer labeling and more consistent comparisons between experimental groups. The figure captions should be more detailed to help the reader understand the key findings.
We have worked on this.
2.2 The experiment involving J54 and GLPG should be described in more detail, particularly regarding the doses and treatment schedules. It would also be helpful to include additional data on toxicity or adverse effects, even if they were minimal.
OK
- Discussion:
3.1 The discussion section briefly summarizes the key findings, but it could be expanded to include a more detailed interpretation of how these results fit within the broader context of cancer metastasis research. Specifically, the authors should explore the potential implications of targeting the TLK1-MK5 axis in combination with other therapies.
See discussion of comment 3
3.2 The experimental validation of the molecular mechanisms underlying TLK1-MK5-mediated metastasis is currently lacking. The authors should propose specific in vitro and in vivo experiments to explore the exact molecular interactions that promote metastatic behavior in PCa.
We have worked on this.
3.3 The manuscript would benefit from a more thorough discussion of the limitations of the study. For example, while the use of the TRAMP mouse model is valuable, it would be helpful to discuss the generalizability of these findings to human cancers and other animal models.
Many researchers don’t like the TRAMP model as it is based on T-Ag transformation and not typical human oncogenes, but it is far simpler than other and those that require the multiple crosses with CRE hosts.
Reviewer 3 Report
Comments and Suggestions for Authors
Manuscript ID: cancers-3531639
Title: Targeting the TLK1-MK5 axis suppresses prostate cancer metastasis
The present study by Olatunde et al., presents a compelling comparison between GLPG and J54 in metastatic progression and tumor growth, highlighting GLPG’s role in reducing metastasis without affecting tumor size, whereas J54 demonstrates dual effects on both metastasis and proliferation through TLK1>NEK1>YAP1 axis modulation. Notably, J54’s suppression of PD-L1 suggests potential immunomodulatory benefits, warranting further exploration in immunocompetent models. However, key limitations remain, including the lack of detailed mechanistic validation for GLPG’s mode of action, missing pharmacokinetic and toxicity comparisons between the two compounds, and the need for immune profiling of J54-treated tumors to confirm its proposed effects on neoantigen generation.
Below are specific areas requiring further experimentation to improve clarity, coherence, and scientific rigor, ensuring the robustness of the conclusions.
Major Comments:
- The study establishes TLK1 as a key player in PCa progression and metastasis via MK5 activation. However, its role in different PCa subtypes, including castration-resistant prostate cancer (CRPC), needs further evaluation.
- To determine if TLK1 overexpression is an adaptive mechanism to androgen receptor signaling inhibition (ARSI) and whether it is necessary for resistance to anti-androgen therapy, additional experiments are needed to conclude the findings.
- The inhibitory effects of J54 (TLK1 inhibitor) and GLPG (MK5 inhibitor) suggest the TLK1>MK5 axis is crucial for metastasis. However, it remains unclear whether MK5 inhibition alone is sufficient to block PCa progression.
- While TLK1’s involvement in PCa progression is supported, its direct role in metastasis (beyond association with MK5) could be further validated using inducible TLK1 knockdown models in vivo to track metastasis formation dynamically to strengthen the translational aspect of the study.
- MK5 inhibition (GLPG) does not completely replicate TLK1 inhibition. Are there compensatory pathways that could bypass MK5 in PCa progression? This requires further experimentation.
- Since the study is translational, it would be valuable to analyze TLK1 and MK5 expression in clinical prostate cancer tissues, including metastatic lesions, to correlate their expression with patient outcomes.
- The presented data suggest that J54 inhibits proliferation while GLPG mainly reduces metastatic dissemination. However, proliferation and metastasis are interconnected processes. Could it be that GLPG indirectly limits metastatic potential by impairing survival signals rather than direct inhibition of dissemination? Functional apoptosis and cell survival assays under metastatic stress conditions (e.g., anchorage-independent growth) could help clarify this.
- Since TLK1 is implicated in DNA damage response (DDR), would its inhibition with J54 increase DNA damage markers (e.g., γH2AX, p53 stabilization) and mutational burden, potentially leading to immune activation? This could be tested using immunohistochemistry (IHC) or comet assays.
- Assess whether combination therapy (J54 + GLPG) provides synergistic benefits over monotherapy, particularly in metastatic models.
- The discrepancy between lung and bone metastasis models suggests that the tumor microenvironment (TME) modulates TLK1>MK5 activity. Further characterization of how ECM stiffness, immune composition, and vascularity influence metastasis is needed.
- In the tibia model, GLPG-treated cells showed impaired engraftment, while J54-treated cells proliferated efficiently. Investigating TLK1/MK5-dependent gene expression changes in bone-seeking PCa cells could provide more insight into these observations.
- GLPG’s failure to reduce tumor nodule size in the lungs suggests it primarily affects metastatic dissemination rather than proliferation. Time-course analysis of GLPG treatment at different metastatic stages (early vs. late intervention) would clarify its mechanism.
- TLK1 has ~150 targets, raising the concern about unintended downstream effects beyond metastasis inhibition. The authors should discuss these potential off-target effects in more detail.
- The proposed link between TLK1 inhibition and the Nek1>YAP pathway in tumor progression should be directly tested using genetic (CRISPR knockout) or pharmacological (Nek1/YAP inhibitors) approaches. This is particularly important since YAP signaling is a major regulator of cancer cell plasticity, therapy resistance, and metastatic potential.
- CRISPR/Cas9-mediated knockout of TLK1 and/or MK5 in PCa cells should be performed to confirm that metastatic potential is TLK1>MK5-dependent.
- Use of rescue experiments (reintroducing wild-type or phospho-mutant MK5) will clarify if MK5 phosphorylation is essential for metastatic dissemination.
- Analyzing publicly accessible single-cell transcriptomics of lung and bone metastatic PCa cells can reveal whether different transcriptional programs are activated based on the metastatic microenvironment.
- Investigating whether TLK1/MK5 activation enhances cancer stem cell-like properties using sphere formation assays and ALDH+ stem cell marker analysis would provide valuable insights into the metastatic potential.
- Does TLK1 inhibition sensitizes PCa cells to ARSI or chemotherapy (e.g., docetaxel), which could inform future combination therapy strategies.
- Given that TLK1 and MK5 regulate cytoskeletal dynamics, their role in modulating immune evasion and interactions with tumor-associated macrophages (TAMs) or T cells should be tested. Flow cytometry or single-cell RNA-seq of immune populations in the metastatic microenvironment could provide insight.
Minor Comments:
- The introduction effectively sets the stage for the study, but some sentences are overly complex. Consider breaking them into shorter, more digestible statements for better readability. Example: The first sentence could be revised to: "Prostate cancer (PCa) progresses slowly, particularly when managed with androgen deprivation therapy (ADT). However, metastases still account for approximately 30,000 deaths per year in the U.S”.
- Lines 31-33 - The survival times for metastases to different organs are useful, but it would be helpful to provide more context on how these numbers were derived.
- The statement "TLK involvement in cellular motility is an understudied area." is important. Consider briefly mentioning why previous studies have not focused on this aspect.
- Lines 47-51 - The reference to TLK2 enhancing migration in breast cancer and glioblastoma is interesting. A sentence comparing how TLK1 and TLK2 may differ in function would add value.
- Lines 52-58 - The link between ADT, TLK1B, and mTOR is well-stated, but explaining how TLK1B differs from TLK1 in function would strengthen this argument.
- Ensure that citation [1] refers to the most up-to-date and relevant dataset.
- Some minor errors (e.g., "confoirm" should be "confirm") should be corrected.
- Some sentences are long and complex. Breaking them into two or simplifying them would improve readability
- Some sentences are long and complex. Breaking them into two or simplifying them would improve readability
Author Response
Reviewer 3
The present study by Olatunde et al., presents a compelling comparison between GLPG and J54 in metastatic progression and tumor growth, highlighting GLPG’s role in reducing metastasis without affecting tumor size, whereas J54 demonstrates dual effects on both metastasis and proliferation through TLK1>NEK1>YAP1 axis modulation. Notably, J54’s suppression of PD-L1 suggests potential immunomodulatory benefits, warranting further exploration in immunocompetent models. However, key limitations remain, including the lack of detailed mechanistic validation for GLPG’s mode of action, missing pharmacokinetic and toxicity comparisons between the two compounds, and the need for immune profiling of J54-treated tumors to confirm its proposed effects on neoantigen generation.
We have these but it will have to be in the next publication. GLPG ‘can’ inhibit also MK2, so it may not be strictly specific for MK5. But we did report that it strongly inhibits motility and invasion in all the PCa cell lines we tested with various Incucyte tests, in most cells by altering the cytoskeletal organization and their morphology.
Below are specific areas requiring further experimentation to improve clarity, coherence, and scientific rigor, ensuring the robustness of the conclusions.
Major Comments:
- The study establishes TLK1 as a key player in PCa progression and metastasis via MK5 activation. However, its role in different PCa subtypes, including castration-resistant prostate cancer (CRPC), needs further evaluation.
We chose PC3 (AR- and Androgen-insensitive) as currently the best available model. Most other cell lines and PDX lines do not metastasize efficiently.
- To determine if TLK1 overexpression is an adaptive mechanism to androgen receptor signaling inhibition (ARSI) and whether it is necessary for resistance to anti-androgen therapy, additional experiments are needed to conclude the findings.
We have future plans to cross the TRAMP mice to TLK1-KO (available from Travis Stracker). It will take months and funding which we do not have.
- The inhibitory effects of J54 (TLK1 inhibitor) and GLPG (MK5 inhibitor) suggest the TLK1>MK5 axis is crucial for metastasis. However, it remains unclear whether MK5 inhibition alone is sufficient to block PCa progression.
Only clinical trials will be conclusive on this issue. However, GLPG alone pre-treatment of PC3 cell was sufficient to impede their grafting into the tibia.
- While TLK1’s involvement in PCa progression is supported, its direct role in metastasis (beyond association with MK5) could be further validated using inducible TLK1 knockdown models in vivo to track metastasis formation dynamically to strengthen the translational aspect of the study.
As stated above, we have plans to use TRAMPXTLK1-KO crosses in the future. There are also plans to use PTEN-KO models that should result in ‘increased’ expression of TLK1B via mTOR. We do not yet have funds for such studies, even though they have been planned for years.
- MK5 inhibition (GLPG) does not completely replicate TLK1 inhibition. Are there compensatory pathways that could bypass MK5 in PCa progression? This requires further experimentation.
Likely the TMPRSS2-ERG and TLK1>NEK1>YAP pathways can compensate in part for the direct inhibition of MK5 alone.
- Since the study is translational, it would be valuable to analyze TLK1 and MK5 expression in clinical prostate cancer tissues, including metastatic lesions, to correlate their expression with patient outcomes.
In all our studies, the subjects are de-identified, and we do not have the much more complex IRB approvals for prospective or even retrospective studies. At this time, we can only make correlative diagnostic correlations. The TLK1 and pMK5 antisera are available to all clinical investigators and Pathologists and we hope that someone will try to corroborate our work.
- The presented data suggest that J54 inhibits proliferation while GLPG mainly reduces metastatic dissemination. However, proliferation and metastasis are interconnected processes. Could it be that GLPG indirectly limits metastatic potential by impairing survival signals rather than direct inhibition of dissemination? Functional apoptosis and cell survival assays under metastatic stress conditions (e.g., anchorage-independent growth) could help clarify this.
We published some of this in our 2 mechanistic studies. At the GLPG doses we used, there was no evidence of growth inhibition nor apoptosis. There was near complete loss of motility and invasion through different ECM matrices.
https://doi.org/10.1002/1878-0261.13183 and https://doi.org/10.3390/cancers14235728
- Since TLK1 is implicated in DNA damage response (DDR), would its inhibition with J54 increase DNA damage markers (e.g., γH2AX, p53 stabilization) and mutational burden, potentially leading to immune activation? This could be tested using immunohistochemistry (IHC) or comet assays.
It does, but this will be the subject of our next paper. But also, please see this for the first part of the sentence https://pubmed.ncbi.nlm.nih.gov/23946870/ and https://pubmed.ncbi.nlm.nih.gov/38001987/
- Assess whether combination therapy (J54 + GLPG) provides synergistic benefits over monotherapy, particularly in metastatic models.
The inhibition of metastasis via tail vein was quite substantial (Fig. 3). To see if there would be synergistic or additive effects with a combination therapy (even at only one dose of each), would require a very large number of mice and additional experiments (long if we wanted to proceed beyond the use of IVIS).
- The discrepancy between lung and bone metastasis models suggests that the tumor microenvironment (TME) modulates TLK1>MK5 activity. Further characterization of how ECM stiffness, immune composition, and vascularity influence metastasis is needed.
There are no significant immune components in these NOD-SCID mice. As most of the pre-treated PC3_luc cells did not ‘take/engraft’ (despite the substantial inoculum, I am not sure what information could be gleaned from investigating the ECM stiffness in the trabecular niche. It is a specific property of the pre-treated cells, which is why we treated the cells in vitro before injection, and not the mice.
- In the tibia model, GLPG-treated cells showed impaired engraftment, while J54-treated cells proliferated efficiently. Investigating TLK1/MK5-dependent gene expression changes in bone-seeking PCa cells could provide more insight into these observations.
We did carry out experiment to look at some genes involved in motility/metastasis, post-transcriptionally – neither TLK1 nor MK5 are known to be direct effectors of transcription.
- GLPG’s failure to reduce tumor nodule size in the lungs suggests it primarily affects metastatic dissemination rather than proliferation. Time-course analysis of GLPG treatment at different metastatic stages (early vs. late intervention) would clarify its mechanism.
We did show the IVIS result after just 1 week ‘early on’ and then collected the lungs and analyzed the tumor nodules one month later (late intervention). I am not sure what any intermediate time point would reveal more than that.
- TLK1 has ~150 targets, raising the concern about unintended downstream effects beyond metastasis inhibition. The authors should discuss these potential off-target effects in more detail.
How? TLK1-KO mice have very subtle phenotypes – only under very special conditions. They seem mostly healthy and fine. What should we look for that the investigators that made the mice did not find? https://pubmed.ncbi.nlm.nih.gov/28708136/
- The proposed link between TLK1 inhibition and the Nek1>YAP pathway in tumor progression should be directly tested using genetic (CRISPR knockout) or pharmacological (Nek1/YAP inhibitors) approaches. This is particularly important since YAP signaling is a major regulator of cancer cell plasticity, therapy resistance, and metastatic potential.
We did: both NEK1-KO via CRISPR and inhibition of NEK1>YAP with J54
NEK1 Phosphorylation of YAP Promotes Its Stabilization and Transcriptional Output - PubMed and TLK1>Nek1 Axis Promotes Nuclear Retention and Activation of YAP with Implications for Castration-Resistant Prostate Cancer - PubMed
- CRISPR/Cas9-mediated knockout of TLK1 and/or MK5 in PCa cells should be performed to confirm that metastatic potential is TLK1>MK5-dependent.
We used MK5-KO cells and cells with the MK5-S354A replacement (major TLK1 target residue) to show their immobilization and loss of invasion
https://doi.org/10.1002/1878-0261.13183
- Use of rescue experiments (reintroducing wild-type or phospho-mutant MK5) will clarify if MK5 phosphorylation is essential for metastatic dissemination.
We did – see response above.
- Analyzing publicly accessible single-cell transcriptomics of lung and bone metastatic PCa cells can reveal whether different transcriptional programs are activated based on the metastatic microenvironment.
Good idea but not critical for this paper.
- Investigating whether TLK1/MK5 activation enhances cancer stem cell-like properties using sphere formation assays and ALDH+ stem cell marker analysis would provide valuable insights into the metastatic potential.
Similar studies were already carried out with PC3 cells, and they have a very low proportion of one would qualify as stem cells. We do not think that such properties have much to do with our current study.
https://doi.org/10.3892/ol.2012.1090
- Does TLK1 inhibition sensitizes PCa cells to ARSI or chemotherapy (e.g., docetaxel), which could inform future combination therapy strategies.
Yes. It confers high synthetic lethality with either ARSI or cisplatin.
https://doi.org/10.1002/ijc.32200 and https://doi.org/10.3390/biomedicines11112987
- Given that TLK1 and MK5 regulate cytoskeletal dynamics, their role in modulating immune evasion and interactions with tumor-associated macrophages (TAMs) or T cells should be tested. Flow cytometry or single-cell RNA-seq of immune populations in the metastatic microenvironment could provide insight.
We have already done some of this, but it will need to be the next paper that deals with an immunocompetent syngeneic model that is very different from those presented in this current work.
Minor Comments:
- The introduction effectively sets the stage for the study, but some sentences are overly complex. Consider breaking them into shorter, more digestible statements for better readability. Example: The first sentence could be revised to: "Prostate cancer (PCa) progresses slowly, particularly when managed with androgen deprivation therapy (ADT). However, metastases still account for approximately 30,000 deaths per year in the U.S”.
- Lines 31-33 - The survival times for metastases to different organs are useful, but it would be helpful to provide more context on how these numbers were derived.
- The statement "TLK involvement in cellular motility is an understudied area." is important. Consider briefly mentioning why previous studies have not focused on this aspect.
- Lines 47-51 - The reference to TLK2 enhancing migration in breast cancer and glioblastoma is interesting. A sentence comparing how TLK1 and TLK2 may differ in function would add value.
- Lines 52-58 - The link between ADT, TLK1B, and mTOR is well-stated, but explaining how TLK1B differs from TLK1 in function would strengthen this argument.
- Ensure that citation [1] refers to the most up-to-date and relevant dataset.
- Some minor errors (e.g., "confoirm" should be "confirm") should be corrected.
- Some sentences are long and complex. Breaking them into two or simplifying them would improve readability
We are very appreciative of the reviewer’s suggestions in the Minor comments regarding editing issues.
Round 2
Reviewer 1 Report
Comments and Suggestions for Authors
The authors have addressed the comments I raised, and the figures have been updated as recommended.
Author Response
Dear Dr. Zhang. Thanks for noticing that the self-citation was still excessive. I have now hopefully corrected the problem with 9 out 70 citations. We are really appreciative of the excellent reviews and guidance we have received all along.
Reviewer 2 Report
Comments and Suggestions for Authors
After carefully reading the revised manuscript, I think the authors have largely responded to the main issues I raised in the initial review and have made significant improvements in the experimental design, dosage instructions, figure annotations, and discussion sections. The description of experimental details, statistical methods, and model selection in the revised manuscript is more detailed. However, there are still some details in the manuscript that need further improvement, and the authors are requested to pay attention to and correct them before the final manuscript is finalized.
General Comments:
- The authors have extended their discussion on the role of the TLK1-MK5 axis in prostate cancer metastasis and its potential interactions with other signaling pathways, and have also provided some explanations on the use of the TRAMP model. However, the discussion of the limitations of the TRAMP model in reflecting human prostate cancer metastasis is not sufficient. It is recommended that the authors further clarify the possible gap between the model and clinical practice, and explore how to verify the feasibility of the results of this study in the future by expanding the sample size or multi-center collaboration.
Specific Comments
- Although the manuscript has detailed descriptions of the statistical test methods and p-values ​​used, the descriptions of effect size, confidence intervals, and outlier handling are still somewhat insufficient. It is recommended that the authors further supplement these contents to enhance the rigor and persuasiveness of the data analysis.
- Although the discussion section has been significantly expanded, the authors are still advised to more clearly point out the possible impact of the small sample size of some experiments on the research conclusions in the discussion, and look forward to the possibility of verifying the results of this study by expanding the sample size or multi-center collaboration in the future.
Overall, the revised manuscript has been significantly improved in terms of content completeness, structural clarity, and academic rigor, and most of the initial review comments have been effectively responded to. Nevertheless, there are still some issues that need further revision, especially the discussion section needs further improvement. It is recommended that the author make minor revisions to the above specific comments, and the manuscript will meet the requirements for publication.
Author Response
General Comments:
- The authors have extended their discussion on the role of the TLK1-MK5 axis in prostate cancer metastasis and its potential interactions with other signaling pathways, and have also provided some explanations on the use of the TRAMP model. However, the discussion of the limitations of the TRAMP model in reflecting human prostate cancer metastasis is not sufficient. It is recommended that the authors further clarify the possible gap between the model and clinical practice, and explore how to verify the feasibility of the results of this study in the future by expanding the sample size or multi-center collaboration.
We have further explained the limitations and concerns about the use of the TRAMP model in relation to its extrapolation for human PCa metastases. We have also proposed as a possible alternative and future model the use of PCa-targeted PTEN-KO, which however, requires a more complex breeding strategy before crossing to MK5-KO. We have also added some more discussion of the Tibia inoculation and its implications.
Specific Comments
- Although the manuscript has detailed descriptions of the statistical test methods and p-values ​​used, the descriptions of effect size, confidence intervals, and outlier handling are still somewhat insufficient. It is recommended that the authors further supplement these contents to enhance the rigor and persuasiveness of the data analysis.
We have further expanded the description of the statistical analysis and handling of the outliers both in the Results and M&M sections.
- Although the discussion section has been significantly expanded, the authors are still advised to more clearly point out the possible impact of the small sample size of some experiments on the research conclusions in the discussion, and look forward to the possibility of verifying the results of this study by expanding the sample size or multi-center collaboration in the future.
Yes, we are aware that the conclusions were based on relatively small sample sizes and that despite the support of convincing statistical analyses they will require the verification of multi-center collaboration studies before applying for clinical trials.